# Anatomy-guided Test-Time Adaptation for Automated Fetal Brain MRI Morphometry

**Yijin Li**[*1]                                    LIYIJIN6815@BUAA.EDU.CN
**Mingxuan Liu**[*2]                                    ARKTISX@FOXMAIL.COM
**Hongjia Yang**[2]                              YANGHJ23@MAILS.TSINGHUA.EDU.CN
**Haoxiang Li**[2]                              LIHAOXIA24@MAILS.TSINGHUA.EDU.CN
**Xuguang Bai**[2]                                BXG21@MAILS.TSINGHUA.EDU.CN
**Yi Liao**[3]                                      CONNIE0064@126.COM
**Haibo Qu**[3]                                      WINDOWSQHB@126.COM
**Qiyuan Tian**[†2]                                QIYUANTIAN@TSINGHUA.EDU.CN

[1] *School of Biological Science and Medical Engineering, Beihang University*

[2] *School of Biomedical Engineering, Tsinghua University*

[3] *Department of Radiology, West China Second University Hospital, Sichuan University*

**Editors:** Accepted for publication at MIDL 2025

## Abstract

Fetal brain MRI enables prenatal diagnosis of neurological and developmental diseases through linear morphologic measurements. Traditional manual measurements derived from the visual assessment of 2D MRI slices is labor-intensive, expertise-dependent, and suffers from high inter- and intra-rater variability due to inconsistent slice selection. Deep learning-based automated fetal brain MRI morphometry has been proposed to address these limitations. However, these automated models still struggle with limited generalizability due to cross-device MRI variability and lesion heterogeneity. To solve the problem, we propose an anatomy-guided test-time adaptation (TTA) method integrating a local-global dual-network with anatomical priors via atlas registration, enhancing cross-domain adaptability.

**Keywords:** Fetal brain MRI, Landmark detection, Test-time adaption, Morphometry

## 1. Introduction

Fetal MRI provides superior soft tissue contrast, safety, and reduced sensitivity to maternal body habitus (Rutherford et al., 2008; Ebner et al., 2020; Nagaraj and Kline-Fath, 2022), making it vital for prenatal CNS diagnosis (Torrents-Barrena et al., 2019; Liu et al., 2024; Hao et al., 2025). Superresolution 3D MRI reconstruction (SRR) methods (NiftyMIC (Ebner et al., 2020) or NeSVoR (Xu et al., 2023)) enable the generation of 3D volumes from 2D thick slices, facilitating further analyses such as fetal brain morphometry, which can be applied to gestational age estimation and ventriculomegaly diagnosis (She et al., 2023; Vahedifard et al., 2023). However, manual measurements demand specialized expertise and significant time investment (Gholipour et al., 2012), highlighting the importance of deep learning-based morphometry. Recent automated pipelines combine mask segmentation, 3D

---

[*] Contributed equally

[†] Corresponding author

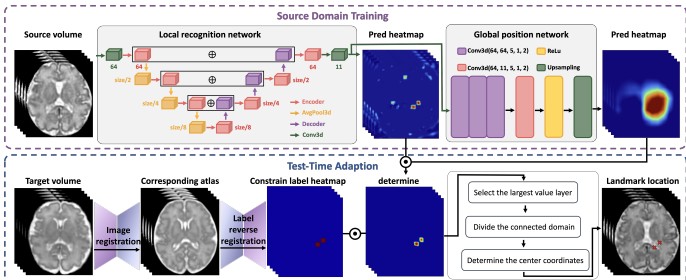

Figure 1: Pipeline of the proposed anatomy-guided TTA method for automated fetal brain MRI morphometry

reconstruction, and CNN-based landmark detection (Masterl et al., 2025; Luis et al., 2025). However, variations in scanning protocols across institutions limit model generalizability to out-of-distribution data (Faghihpirayesh et al., 2024). To address the problem, we propose an anatomy-guided test-time adaptation (TTA) method for fetal brain MRI morphometry. After training a landmark localization model on our private dataset, we apply TTA to the public FeTA 2022 dataset (Payette et al., 2025) using atlas label registration to impose anatomical priors on predicted heatmaps. Experimental results confirm our TTA method enhances measurement accuracy.

## 2. Methods

### 2.1. Data Description

This study used data from 71 pregnant women with normal fetuses, acquired using a Philips Achieva scanner at the West China Second University Hospital of Sichuan University (WCSUH-SCU), as the source domain dataset for model training. Ten cases from the FeTA 2022 (Payette et al., 2025) dataset were used as the target domain test dataset to evaluate the model's generalization performance on cross-device MRI data. Following clinical guidelines, 11 pairs of anatomical landmarks (Garel, 2004) were selected and measured on the corresponding anatomical planes. Specifically: RLV_A (Right Lateral Ventricle_Axial), LLV_A (Left Lateral Ventricle_Axial), RLV_C (Right Lateral Ventricle_Coronal), LLV_C (Left Lateral Ventricle_Coronal), BBD (Bone Biparietal Diameter), CBD (Cerebral Biparietal Diameter), TCD (Transverse Cerebellar Diameter), FOD (Fronto-Occipital Diameter), AVD (Anteroposterior Vermis Diameter), VH (Vermis Height), and CCL (Corpus Callosum Length). The annotations for all data were manually labeled by an experienced radiologist (Y.L.) using ITK-SNAP (Yushkevich et al., 2016).

### 2.2. Source Domain Training

For training on the source domain dataset, as shown in Figure 1, we used 3D-SCN (Payer et al., 2019) consists of two components: a Local Recognition Network (LRN) and a Global Positioning Network (GPN). The LRN generates multiple candidate regions with high

heatmap responses, while the GPN applies convolutions with larger receptive fields to constrain the potential spatial range of target locations.

### 2.3. Test-Time Adaption

As shown in Figure 1, our anatomy-guided TTA method constrains heatmap predictions using atlas registration. The test image is registered to an age-specific fetal brain template, and template labels are inversely transformed to the test space, creating anatomical constraint regions. Within these regions, landmarks are localized on slices with maximum heatmap response. This approach combines the LRN's candidate regions with the GPN's spatial constraints, enhanced by anatomical priors for improved cross-device adaptability. Biometric measurements are calculated as distances between the localized landmark pairs.

### 3. Results and Conclusion

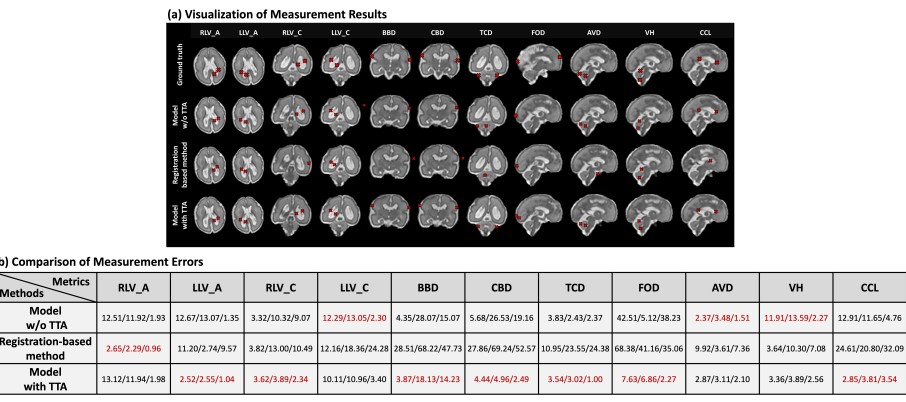

**(a) Visualization of Measurement Results**

**(b) Comparison of Measurement Errors**

| Metrics / Methods | RLV_A | LLV_A | RLV_C | LLV_C | BBD | CBD | TCD | FOD | AVD | VH | CCL |
|---|---|---|---|---|---|---|---|---|---|---|---|
| Model w/o TTA | 12.51/11.92/1.93 | 12.67/13.07/1.35 | 3.32/10.32/9.07 | 12.29/13.05/2.30 | 4.35/28.07/15.07 | 5.68/26.53/19.16 | 3.83/2.43/2.37 | 42.51/5.12/38.23 | 2.37/3.48/1.51 | 11.91/13.59/2.27 | 12.91/11.65/4.76 |
| Registration-based method | 2.65/2.29/0.96 | 11.20/2.74/9.57 | 3.82/13.00/10.49 | 12.16/18.36/24.28 | 28.51/68.22/47.73 | 27.86/69.24/52.57 | 10.95/23.55/24.38 | 68.38/41.16/35.06 | 9.92/3.61/7.36 | 3.64/10.30/7.08 | 24.61/20.80/32.09 |
| Model with TTA | 13.12/11.94/1.98 | 2.52/2.55/1.04 | 3.62/3.89/2.34 | 10.11/10.96/3.40 | 3.87/18.13/14.23 | 4.44/4.96/2.49 | 3.54/3.02/1.00 | 7.63/6.86/2.27 | 2.87/3.11/2.10 | 3.36/3.89/2.56 | 2.85/3.81/3.54 |

Figure 2: **Comparison Results.** (a) Visualization of measurement Results. (b) Table of measurement errors (Average Starting Point Error / Average Ending Point Error / Average Measurement Error, all measured in $mm$).

We conducted experiments on the FeTA dataset to evaluate three methods: i) testing the model without TTA, ii) the registration-based method, and iii) testing the model with TTA. As Figure 2 shows, the testing results of model without TTA exhibited localization deviations at boundary landmarks, indicating limited adaptability in low-contrast regions. The registration-based method underperformed in ventriculomegaly cases due to registration sensitivity. Our proposed TTA method achieved the lowest error in seven out of eleven biometric measurements. In conclusion, the proposed anatomy-guided TTA method effectively enhances fetal brain MRI morphometry across different scanning devices by integrating local-global neural networks with anatomical prior constraints.

### Acknowledgments

Tsinghua University Startup Fund and Dushi Program (grant number 20241080026).

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
