# OpenReview forum: "Anatomy-guided Test-Time Adaptation for Automated Fetal Brain MRI Morphometry"
_MIDL.io/2025/Short_Papers — MIDL 2025 - Short Papers_

### Official Review · Reviewer_bogz · 2025-04-28

**Rating:** 4
**Confidence:** 5

**Summary:**

This paper introduces an anatomy-guided test-time adaptation (TTA) strategy aimed at improving the generalization of automated fetal brain MRI morphometry across different imaging devices and scanning protocols. The authors train a local-global dual-network model on a private dataset and apply TTA during testing on the public FeTA 2022 dataset. Experimental results demonstrate that incorporating anatomical priors using atlas label registration through TTA significantly improves landmark localization compared to both a baseline model without adaptation and a registration-only method.

**Strengths:**

- The challenge of cross-domain generalization in fetal brain MRI is rarely explored but is critically important for clinical practice.

- Integrating anatomical priors during TTA offers a distinct approach compared to methods that require retraining or domain-specific fine-tuning.

**Weaknesses:**

- The testing dataset is quite limited (only 10 cases from the FeTA dataset) for a thorough evaluation.

- The baseline comparisons are restricted to a simple registration-based method and a model without TTA. Including comparisons with existing domain adaptation or TTA methods would provide a clearer understanding of the contribution and effectiveness of the proposed approach.

- Additionally, providing more technical details on the atlas-based image registration process and the label reverse registration used to obtain anatomical priors would further improve the clarity and reproducibility of the method.

---

### Decision · Program_Chairs · 2025-05-01

Accept